# Unlocking Kuhn Verdazyls: New Synthetic Approach and Useful Mechanistic Insights

**DOI:** 10.3390/ijms24032693

**Published:** 2023-01-31

**Authors:** Fedor E. Teslenko, Leonid L. Fershtat

**Affiliations:** Laboratory of Nitrogen Compounds, N.D. Zelinsky Institute of Organic Chemistry, Russian Academy of Sciences, Leninsky Prosp., 47, 119991 Moscow, Russia

**Keywords:** polynitrogen heterocycles, tetrazines, Kuhn verdazyls, cyclization, organic radicals

## Abstract

An optimized synthetic protocol toward the assembly of Kuhn verdazyls based on an azo coupling of arenediazonium salts with readily available hydrazones followed by the base-mediated cyclization of in situ formed formazans with formalin was developed. The scope and limitations of the presented method were revealed. Some new mechanistic insights on the formation of Kuhn verdazyls were also conducted. It was found that in contradiction with previously assumed hypotheses, the synthesis of verdazyls was accomplished via an intermediate formation of verdazylium cations which were in situ reduced to leucoverdazyls. The latter underwent deprotonation under basic conditions to generate corresponding anions which coproportionate with verdazylium cations to furnish the formation of Kuhn verdazyls. The spectroscopic and electrochemical behavior of the synthesized verdazyls was also studied. Overall, our results may serve as a reliable basis for further investigation in the chemistry and applications of verdazyls.

## 1. Introduction

A creation of novel functional organic materials remains one of the urgent goals in modern chemistry and materials science [1,2,3,4]. Such materials constitute a large variety of usually conjugated organic compounds with different chemical and physical properties. Recent achievements of numerous research groups worldwide confirmed that an incorporation of a nitrogen heteroaromatic motif usually enhances the quality of materials compared to their carbocyclic analogues [5,6,7]. Nitrogen heterocycles are often incorporated in a preparation of novel pharmacologically active drug candidates and therapeutic agents [8,9,10]. Polynitrogen heterocyclic scaffolds are also valuable structural subunits in a construction of functional materials for various energetic applications [11,12,13,14,15].

Among promising functional organic materials, stable organic radicals have long been of fundamental interest and now are employed as spin labels [16,17] in supramolecular chemistry and as components of magnetic and conducting organic materials [18]. From the synthetic point of view, stable radicals possess unique traits and participate in uncommon reactions providing different exotic structures which could not be obtained through other synthetic strategies. In a series of stable radicals, nitrogen heterocyclic radicals usually possess optimized functional properties and may be incorporated in a number of devices [19,20,21,22]. In this regard, verdazyls are one of the known stable heterocycle-based nitrogen-centered organic radicals incorporating a partially saturated tetrazine ring. The first representatives of verdazyls were synthesized by Richard Kuhn in the 1960s [23]. Nowadays, two main types of verdazyl derivatives—Kuhn verdazyls and oxoverdazyls—are known (Figure 1).

The synthesis and reactivity of oxoverdazyls along with their practicability were well established in recent years [24,25,26,27]. On the other hand, investigations regarding Kuhn verdazyls are substantially rare. Kuhn verdazyls may serve as ligands in a preparation of metal–organic complexes [28] or as substrates for the electrochemical generation of carbon-centered radicals [29,30]. Recent in-depth studies revealed the utility of Kuhn verdazyls as donor molecules in electron transfer reactions and their capacity for weak antiferromagnetic coupling [31] as well as their application as components of redox flow batteries [32]. A common method for the preparation of Kuhn verdazyls is based on the methylation of the tailor-made formazans followed by aerial oxidation (Figure 1a) [33,34,35]. However, this method has a number of limitations including narrow substrate scope and requires a necessary isolation of formazans which are well-known inconveniently isolable dyes. Moreover, the mechanism of the verdazyl formation is still not well understood; commonly, it is assumed that the main reaction step is carried out by the oxidation of intermediate leucoverdazyls with oxygen. Hence, we were interested in the development of an optimized synthetic protocol toward the assembly of Kuhn verdazyls based on an azo coupling of arenediazonium salts with readily available hydrazones followed by the base-mediated cyclization of in situ formed formazans with formalin (Figure 1b). Some new mechanistic insights on the formation of Kuhn verdazyls were also conducted.

## 2. Results and Discussion

Our investigations toward the desired synthetic protocol started from the optimization of reaction conditions. Hydrazone **1a** and *p*-tolyl diazonium salt were used as model substrates (Table 1). Initially, we tried to employ a known approach for the synthesis of formazan precursors using in situ generated diazonium salt; however, some test reactions leading to arylformazan **2a** formation suffered from low yields (entries 1–3). In addition, we failed to perform the cyclization of thus obtained formazan **2a** either with paraform or formalin to the target verdazyl **3a**. A replacement of DMF with DMSO was also ineffective (entry 4). Next, preliminary isolated diazonium salt was used instead of it being generated in situ, albeit no formazan formation was observed without base (entry 5), so NaOH and pyridine were applied (entries 6, 7). Inspired by these results, methylation was conducted with formaline as formaldehyde source without formazan isolation, since it was assumed that the main loss of substance occurred upon isolation. This attempt resulted in a formation of the desired verdazyl **3a** in a good yield (entry 9). Several control experiments were conducted to ensure the role of NaOH on the methylation step (entry 10) and preliminary isolated diazonium salt (entry 11), and no verdazyl formation was detected as expected.

With optimized conditions in hand, we applied our method on a series of hydrazones and diazonium salts, and a scope of Kuhn verdazyls was obtained (Figure 2). The developed synthetic protocol well tolerated various substituents onto aromatic rings in both substrates. In general, diverse verdazyls **3a-k** bearing various alkyl-, alkoxy- and halogen substituents onto the phenyl ring were obtained in high yields (67–82%). However, our approach had some limitations: we failed to synthesize verdazyls **3l,m** bearing a high electron-withdrawing nitro group and verdazyls **3n,o** incorporating 3-chlorophenyl moiety at the nitrogen atom. For comparison, the same motif at the carbon atom in starting hydrazone well tolerated the investigated protocol, and the corresponding verdazyl **3g** was obtained in a good yield. Interestingly, an incorporation of the 3-pyridyl subunit also resulted in a formation of verdazyl **3i**, which additionally confirms the versatility of our approach. In addition, our method was found to be scalable, which was proven by the Gram-scale synthesis of representative verdazyl **3a** (yield 5.12 g, 75%). The structure and composition of verdazyls **3a-k** were confirmed by IR and EPR spectroscopy, high-resolution mass spectrometry and elemental analysis. The structure of compound **3a** was unambiguously confirmed by single crystal X-ray diffraction data (Figure 2). It should be emphasized that our newly developed approach has a number of advantages in comparison with the known protocols [33,34,35] for the assembly of Kuhn verdazyls, including a one-pot procedure which does not require preliminary isolation of the intermediate formazans, mild reaction conditions and good scalability.

To obtain insight into the mechanism of verdazyl formation, several analytical studies and control experiments were conducted. As it was mentioned above, conventionally verdazyls are produced via the aerial oxidation of leucoverdazyls (**LVZ**) [33,34,35] (Figure 3a). However, our findings and further experiments (see below) suggested a more complex mechanistic pathway. The base-mediated deprotonation of formazan **2** followed by the addition of formaldehyde provided *N*-hydroxymethylformazan, which is prone to cyclization to verdazylium cation (**VDC**). The latter could be reduced by formaldehyde in an alkaline environment to provide leucoverdazyl (**LVZ**). Thus, the formed **LVZ** could be deprotonated to provide a corresponding anion (**dpLVZ**), which is prone to coproportionation with verdazylium cation (**VDC**) producing verdazyl **3** (Figure 3b).

To prove this point, we conducted the reaction under argon atmosphere (Figure 4), and verdazyl **3a** was obtained in an 80% yield, which is almost the same in comparison to the reaction performed under aerial conditions. Therefore, the impact of oxygen on the verdazyl formation can be completely neglected. This experiment also confirms that aerial oxidation is not a necessary reaction step in this process.

Moreover, according to the previously studied electrochemical properties of verdazyls, they endure both electrochemically reversible oxidation and reduction processes [36]. It was shown that the oxidation of model verdazyl **3a** to the corresponding cation **VDC** occurred at −0.29 V (vs Fc/Fc^+^), while reduction appeared at −1.27 V (vs. Fc/Fc^+^) (half-reaction potentials are given) and verdazyl **3a** was stable in the potentials between. These data also prove that in situ formed **VDC** species can act as oxidizing agent for the corresponding anion **dpLVZ**, which leads to the coproportionation to verdazyl (Figure 5). These findings are in agreement with a recently reported application of Kuhn verdazyls as components of redox flow batteries based on the same electrochemicaly induced redox processes [32].

We also conducted spectroscopic analytical studies of the synthesized verdazyls under range of pH (5.21–7.00) using UV-Vis spectroscopy. So, in a neutral solution of model verdazyl **3a**, two absorption maxima were observed at 440 nm and 728 nm (Figure 3). The appearance of a new band at 560 nm indicated the formation of a protonated verdazyl derivative raised from the protonation of the nitrogen atom adjacent to the radical center. At lower pH values achieved via the addition of concentrated sulfuric acid, peaks of verdazyl **3a** decreased and a peak at 560 nm increased, confirming the observed changes.

## 3. Materials and Methods

### 3.1. General

All reactions were carried out in well-cleaned oven-dried glassware with magnetic stirring. The IR spectra were recorded on a Bruker “Alpha” spectrometer in the range 400–4000 cm^−1^ (resolution 2 cm^−1^). Elemental analyses were performed by the CHN Analyzer Perkin-Elmer 2400. High-resolution mass spectra were recorded on a Bruker microTOF spectrometer with electrospray ionization (ESI). All measurements were performed in a positive (+MS) ion mode (interface capillary voltage: 4500 V) with a scan range *m/z*: 50–3000. External calibration of the mass spectrometer was performed with Electrospray Calibrant Solution (Fluka). A direct syringe injection was used for all analyzed solutions in MeCN (flow rate: 3 μL min^−1^). Nitrogen was used as the nebulizer gas (0.4 bar) and dry gas (4.0 L min^−1^); the interface temperature was set at 180 °C. All spectra were processed by using the Bruker DataAnalysis 4.0 software package. Analytical thin-layer chromatography (TLC) was carried out on Merck 25 TLC silica gel 60 F_254_ aluminum sheets. The visualization of the TLC plates was accomplished with a UV light. All solvents were purified and dried using standard methods prior to use. All standard reagents were purchased from Aldrich or Acros Organics and used without further purification.

### 3.2. X-ray Crystallographic Data and Refinement Details

X-ray diffraction data were collected at 100 K on a four-circle Rigaku Synergy S diffractometer equipped with a HyPix600HE area-detector (kappa geometry, shutterless ω-scan technique), using graphite monochromatized Cu K_α_-radiation. The intensity data were integrated and corrected for absorption and decay by the CrysAlisPro program [37]. The structure was solved by direct methods using SHELXT [38] and refined on *F^2^* using SHELXL-2018 [39] in the OLEX2 program [40]. All non-hydrogen atoms were refined with individual anisotropic displacement parameters. All hydrogen atoms were placed in ideal calculated positions and refined as riding atoms with relative isotropic displacement parameters (for details, see Appendix A). The Mercury program suite [41] was used for molecular graphics. Deposition number 2,233,690 contains the supplementary crystallographic data. These data are provided free of charge by the joint Cambridge Crystallographic Data Centre.

### 3.3. Synthesis of Verdazyls 3 (General Procedure)

Hydrazone (2 mmol) was dissolved in a mixture of DMF (2 mL) and pyridine (1 mL); then, the reaction mixture was cooled to −5 °C, and diazonium salt (2.2 mmol) was added portionwise. The reaction mixture was stirred at 20 °C for 4 h following by dilution with DMF (23 mL) and addition of the solution of NaOH (0.28 g) in H_2_O (4.5 mL) and 40% formalin solution (2 mL). The reaction was left and stirred overnight; then, Et_2_O (75 mL) was added. The organic phase was separated, washed with H_2_O (9 × 75 mL) and dried over MgSO_4_. Filtration of the drying agent and evaporation of the solvent afforded crude verdazyls **3**, which were purified by recrystallization from MeOH.


**1,5-Di-*p*-tolyl-3-phenylverdazyl (3a)**


Green solid, yield 525 mg (77%). Mp. 108–110 °C. *R*_f_ 0.70 (CHCl_3_). IR (KBr): 2923, 2854, 1656, 1514, 1180, 1146, 816, 694 cm^−1^. EPR (toluene): *g* = 2.0030. HRMS (ESI) Calcd. for C_22_H_21_N_4_ [M]^+^: 341.1761, found: 341.1764. Anal. calcd. for C_22_H_21_N_4_ (%): C, 77.39; H, 6.20; N, 16.41. Found (%): C, 77.45; H, 6.07; N, 16.55.


**1,5-Di-*p*-tolyl-3-(*p*-methoxyphenyl)verdazyl (3b)**


Green solid, yield 571 mg (77%). Mp. 107–109 °C (dec.). *R*_f_ 0.75 (CHCl_3_). IR (KBr): 3030 (m), 2922 (s), 2862 (m), 2838 (m), 1899 (w), 1680 (s), 1610 (s), 1515 (s), 1466 (m), 1250 (s), 1172 (s), 1031 (s), 929 (w), 818 (s) cm^−1^. EPR (toluene): *g* = 2.0035. HRMS (ESI) Calcd. for C_23_H_23_N_4_O [M]^+^: 371.1866, found: 371.1856. Anal. calcd. for C_23_H_23_N_4_O (%): C, 74.37; H, 6.24; N, 15.08. Found (%): C, 74.51; H, 6.09; N, 15.21.


**1,5-Di-*p*-tolyl-3-(*p*-fluorophenyl)verdazyl (3c)**


Green solid, yield 546 mg (76%). Mp. 102–104 °C (dec.). *R*_f_ 0.72 (CHCl_3_). IR (KBr): 3031 (s), 2922 (s), 2864 (m), 1901 (w), 1679 (s), 1610 (s), 1516 (s), 1421 (s), 1156 (s), 1016 (m), 967 (w), 844 (s), 816 (s) cm^−1^. EPR (toluene): *g* = 2.0040. HRMS (ESI) Calcd. for C_22_H_20_FN_4_ [M]^+^: 359.1667, found: 359.1665. Anal. calcd. for C_22_H_20_FN_4_ (%): C, 73.52; H, 5.61; N, 15.59. Found (%): C, 73.39; H, 5.74; N, 15.50.


**1,5-Di-*p*-tolyl-3-(*p*-chlorophenyl)verdazyl (3d)**


Green solid, yield 578 mg (77%). Mp. 103–105 °C. *R*_f_ 0.73 (CHCl_3_). IR (KBr): 3018 (m), 2919 (s), 2860 (m), 1899 (w), 1678 (s), 1610 (s), 1512 (s), 1488 (s), 1249 (m), 1090 (m), 816 (s) cm^−1^. EPR (toluene): *g* = 2.0040. HRMS (ESI) Calcd. for C_22_H_20_^35^ClN_4_ [M]^+^: 375.1371, found: 375.1359. Anal. calcd. for C_22_H_20_ClN_4_ (%): C, 70.30; H, 5.36; N, 14.91. Found (%): C, 70.14; H, 5.22; N, 15.00.


**1,5-Di-*p*-tolyl-3-(*p*-bromophenyl)verdazyl (3e)**


Green solid, yield 655 mg (78%). Mp. 112–114 °C. *R*_f_ 0.73 (CHCl_3_). IR (KBr): 3029 (m), 2921 (s), 2861 (m), 1906 (w), 1686 (s), 1608 (s), 1511 (s), 1179 (m), 1072 (m), 1011 (m), 816 (s) cm^−1^. EPR (toluene): *g* = 2.0030. HRMS (ESI) Calcd. for C_22_H_20_^79^BrN_4_ [M]^+^: 419.0866, found: 419.0868. Anal. calcd. for C_22_H_20_BrN_4_ (%): C, 62.86; H, 4.80; N, 13.33. Found (%): C, 62.99; H, 4.95; N, 13.07.


**1,5-Di-*p*-tolyl-3-(*m*-fluorophenyl)verdazyl (3f)**


Green solid, yield 495 mg (69%). Mp. 100–102 °C (dec.). *R*_f_ 0.78 (CHCl_3_). IR (KBr): 3031 (w), 2922 (m), 2864 (w), 1882 (w), 1683 (m), 1612 (s), 1515 (s), 1456 (m), 1252 (m), 991 (w), 818 (s) cm^−1^. EPR (toluene): *g* = 2.0035. HRMS (ESI) Calcd. for C_22_H_20_FN_4_ [M]^+^: 359.1667, found: 359.1659. Anal. calcd. for C_22_H_20_FN_4_ (%): C, 73.52; H, 5.61; N, 15.59. Found (%): C, 73.66; H, 5.79; N, 15.44.


**1,5-Di-*p*-tolyl-3-(*m*-chlorophenyl)verdazyl (3g)**


Green solid, yield 600 mg (80%). Mp. 106–108 °C (dec.). *R*_f_ 0.75 (CHCl_3_). IR (KBr): 3028 (m), 2921 (s), 2863 (m), 1881 (w), 1679 (s), 1611 (s), 1569 (s), 1250 (s), 1179 (m), 1077 (m), 990 (w), 816 (s) cm^−1^. EPR (toluene): *g* = 2.0040. HRMS (ESI) Calcd. for C_22_H_20_^35^ClN_4_ [M]^+^: 375.1371, found: 375.1377. Anal. calcd. for C_22_H_20_ClN_4_ (%): C, 70.30; H, 5.36; N, 14.91. Found (%): C, 70.37; H, 5.18; N, 15.09.


**1,5-Di-*p*-tolyl-3-(*m*-bromophenyl)verdazyl (3h)**


Green solid, yield 689 mg (82%). Mp. 107–109 °C (dec.). *R*_f_ 0.74 (CHCl_3_). IR (KBr): 3029 (m), 2920 (s), 2863 (m), 1883 (m), 1679 (s), 1611 (s), 1568 (s), 1516 (s), 1250 (s), 1179 (m), 992 (m), 816 (s) cm^−1^. EPR (toluene): *g* = 2.0030. HRMS (ESI) Calcd. for C_22_H_20_^79^BrN_4_ [M]^+^: 419.0866, found: 419.0852. Anal. calcd. for C_22_H_20_BrN_4_ (%): C, 62.86; H, 4.80; N, 13.33. Found (%): C, 62.67; H, 4.52; N, 13.50.


**1,5-Di-*p*-tolyl-3-(3-pyridyl)verdazyl (3i)**


Green solid, yield 506 mg (74%). Mp. 124–126 °C. *R*_f_ 0.47 (CHCl_3_). IR (KBr): 2922 (s), 2853 (m), 1684 (m), 1609 (m), 1512 (s), 1383 (m), 1180 (s), 1077 (s), 816 (m) cm^−1^. EPR (toluene): *g* = 2.0035. HRMS (ESI) Calcd. for C_21_H_20_N_5_ [M]^+^: 342.1713, found: 342.1717. Anal. calcd. for C_21_H_20_N_5_ (%): C, 73.66; H, 5.89; N, 20.45. Found (%): C, 73.50; H, 6.01; N, 20.62.


**1-(*p*-Tolyl)-3-phenyl-5-(ethoxyphenyl)verdazyl (3j)**


Brown solid, yield 527 mg (71%). Mp. 110–112 °C (dec.). *R*_f_ 0.75 (CHCl_3_). IR (KBr): 2956 (m), 2920 (s), 2851 (m), 1682 (w), 1610 (m), 1513 (s), 1258 (m), 1179 (m), 1078 (m), 814 (m) cm^−1^. EPR (toluene): *g* = 2.0040. HRMS (ESI) Calcd. for C_23_H_23_N_4_O [M]^+^: 371.1866, found: 371.1872. Anal. calcd. for C_23_H_23_N_4_O (%): C, 74.37; H, 6.24; N, 15.08. Found (%): C, 74.22; H, 6.13; N, 14.84.


**1,5-Di-(*p*-ethoxyphenyl)-3-(*p*-methoxyphenyl)verdazyl (3k)**


Brown solid, yield 578 mg (67%). Mp. 106–108 °C (dec.). *R*_f_ 0.72 (CHCl_3_). IR (KBr): 2979 (s), 2934 (s), 2839 (m), 1682 (s), 1605 (s), 1512 (s), 1393 (m), 1251 (s), 1172 (s), 1046 (s), 923 (m), 837 (s) cm^−1^. EPR (toluene): *g* = 2.0040. HRMS (ESI) Calcd. for C_25_H_27_N_4_O_3_ [M]^+^: 431.2078, found: 431.2073. Anal. calcd. for C_25_H_27_N_4_O_3_ (%): C, 69.59; H, 6.31; N, 12.98. Found (%): C, 69.77; H, 6.41; N, 12.72.


**Gram-scale procedure for the synthesis of verdazyl 3a**


1-(4-chlorobenzylidene)-2-(*p*-tolyl)hydrazine (4.88 g, 20 mmol) was dissolved in a mixture of DMF (20 mL) and pyridine (10 mL); then, the reaction mixture was cooled to −5 °C and *p*-tolyldiazonium tetrafluoroborate (4.53 g, 22 mmol) was added portionwise. The reaction mixture was stirred at 20 °C for 4 h, which was followed by dilution with DMF (150 mL) and addition of the solution of NaOH (2.8 g) in H_2_O (45 mL) and 40% formalin solution (20 mL). The reaction was left stirred overnight; then, Et_2_O (450 mL) was added. Organic phase was separated, washed with H_2_O (9 × 500 mL) and dried over MgSO_4_. Filtration of the drying agent and evaporation of the solvent afforded verdazyl **3a**, which was recrystallized from MeOH. Yield 5.12 g (75%).


**Synthesis of verdazyl 3a under argon atmosphere**


1-(4-chlorobenzylidene)-2-(*p*-tolyl)hydrazine (488 mg, 2 mmol) was dissolved in the degassed and argonized mixture of DMF (2 mL) and pyridine (1 mL); then, the reaction mixture was cooled to −5 °C and *p*-tolyldiazonium tetrafluoroborate (453 mg, 2.2 mmol) was added portionwise. The reaction mixture was stirred at 20 °C for 4 h, which was followed by dilution with degassed and argonized DMF (23 mL) and addition of the solution of NaOH (0.28 g) in H_2_O (4.5 mL) and 40% formalin solution (2 mL). The reaction was left stirred overnight; then, Et_2_O (75 mL) was added. The organic phase was separated, washed with H_2_O (9 × 75 mL) and dried over MgSO_4_. Filtration of the drying agent and evaporation of the solvent afforded verdazyl **3a**, which was recrystallized from MeOH. Yield 546 mg (80%).

## 4. Conclusions

In summary, a new synthetically useful approach toward the assembly of Kuhn verdazyls was developed. This approach includes an interaction of readily available hydrazones with arenediazonium salts with a subsequent base-mediated cyclization of in situ formed formazans with formalin. It was shown that a utilization of isolated diazonium salts is required to achieve high yields of target Kuhn verdazyls. Mechanistic investigations demonstrated that an interaction of in situ generated formazans with formalin resulted in a cyclization into verdazylium cations which underwent subsequent reduction to leucoverdazyls. Under basic conditions, leucoverdazyls are easily deprotonated to the corresponding anions which coproportionate with verdazylium cations to furnish the formation of Kuhn verdazyls. The spectroscopic and electrochemical behavior of the synthesized verdazyls revealed the presence of verdazylium cations as key intermediates of the studied process. Overall, our results may serve as a reliable foundation for further investigation in the chemistry and applications of verdazyls.

## Data Availability

Data obtained in this project are contained within this article and available upon request from the authors.

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
