# Peer review of "Unlocking Kuhn Verdazyls: New Synthetic Approach and Useful Mechanistic Insights"

_ijms, 2023, doi:10.3390/ijms24032693_

Round 1
Reviewer 1 Report
I read the manuscript ijms-2167972 carefully. It is an interesting and seemingly well-prepared work, but it contains a number of shortcomings. Its subject is relatively rarely described stable free radicals called verdazyls. I cannot agree with the statement in the introduction about medical applications except when they can be used in vitro in medical diagnostics, because their size limits their applications similar to nitroxy radicals, especially the TEMPO type, which are biological spin probes.
The prospects of use as a standard in the ESR technique are also questionable due to the complicated resonance spectrum resulting from interaction with the magnetic nuclei of nitrogen atoms. DPPH - one sharp line, TEMPO - symmetrical triplet. These doubts of mine do not diminish the cognitive value of the submitted work, because in recent years the interest in these compounds has increased enormously, if only because of their use as fluorescent free radical catchers (profluorescence). The Authors focused their attention on a new effective method for the synthesis of Kuhn's verdazyls, while proposing a reaction mechanism and proving that air oxidation is not necessary in the last step.
And here I have the most serious reservations. The mechanism proposed by the Authors is wrong, because the first cyclic product is a biradical or a corresponding bicyclic compound. This error results from too superficial analysis by the authors of the transition from formazan to cyclic product and getting lost in the mechanism. In this case, there remains a new problem with the transition from a biradical to a monoradical without involving oxygen from the air. The only factor may be formalin remaining in balance with its dimer (dioxetane), which has a chance to neutralize free radicals very effectively by transforming into bicyclic dioxirane and this in the aquatic environment turns into glyoxal, which is a by-product and should be relatively easy to detect.
Another test for the correctness of the mechanism proposed by the Authors is to prove the autocatalytic course of the reaction, because the VDC formed in the second step is reused in the transition from dpLVZ to 3.
I ask the Authors to once again carefully analyze their proposal for a mechanism and remove inaccuracies from it, being able to use my suggestions. The next objection is the authors' conclusion about the existence of two acid-base equilibriums, because the presence of ideal isosbetic points (Fig. 3, and in Supporting Information - Section S2) indicate a single equilibrium without the existence of LVZ. In addition, all measurements were made in an acidic environment, so the existence of dpLVZ is impossible, and the transition from LVZ to 3a loses an unpaired electron along the way and from a formal point of view cannot be written as such, similar to the transition 3a to dplVZ. Such a record may refer to red-ox processes and not to acid-base equilibrium.
Authors should further revise their views on this issue. I also have reservations about the experimental part, in which there are seemingly three descriptions of the synthesis, and there is only one with an enlarged scale or supplemented with information about the use of argon atmosphere instead of air. It is surprising that, according to the description, a pure product is obtained and even crystalline (for melting points and XRD analysis) without a crystallization process, while the yields are 70-80%. How were 20-30% of unreacted water-insoluble substrates separated? Authors should also pay attention to this when preparing a corrected version of the article. Concluding, I think that the work is interesting and valuable, but it requires a lot of serious corrections.
Author Response
Please see the attached response.

Reviewer 2 Report
Reviewer’s Comments:
The manuscript “Kuhn verdazyls: new insights into the synthesis and mechanism of formation” is a very interesting work. In this work, an optimized synthetic protocol toward assembly of Kuhn verdazyls based on an azo coupling of arenediazonium salts with readily available hydrazones followed by base-mediated cyclization of in situ formed formazans with formalin was developed. Scope and limitations of the presented method were revealed. Some new mechanistic insights on the formation of Kuhn verdazyls were also conducted. It was found that in contradiction with previously assumed hypothesis, the synthesis of verdazyls was accomplished via an intermediate formation of verdazylium cations which were in situ reduced to leucoverdazyls. The latter underwent deprotonation under basic conditions to generate corresponding anions which coproportionate with verdazylium cations to furnish the formation of Kuhn verdazyls. While I believe this topic is of great interest to our readers, I think it needs major revision before it is ready for publication. So, I recommend this manuscript for publication with major revisions.
1. In this manuscript, the authors did not explain the importance of the Kuhn verdazyls in the introduction part. The authors should explain the importance of Kuhn verdazyls.
2) Title: The title of the manuscript is not impressive. It should be modified or rewritten it.
3) Correct the following statement “Spectroscopic and electrochemical behavior of the synthesized verdazyls were also studied. Overall, our results may serve as a reliable foundation for further investigation in the chemistry and applications of verdazyls”.
4) Keywords: The Kuhn verdazyls is missing in the keywords. So, modify the keywords.
5) Introduction part is not impressive. The references cited are very old. So, Improve it with some latest literature like 10.1016/j.molstruc.2021.131145, 10.3389/fchem.2022.1023316
6) The authors should explain the following statement with recent references, “To get insight into the mechanism of verdazyl formation several analytical studies 102
and control experiments were conducted”.
7) Add space between magnitude and unit. For example, in synthesis “21.96g” should be 21.96 g. Make the corrections throughout the manuscript regarding values and units.
8) The author should provide reason about this statement “We also conducted spectroscopic analytical studies of the synthesized verdazyls under range of pH (5.21-7.00). So, in the neutral solution of model verdazyl 3a, two principal peaks were observed at 440 nm and 728 nm”.
9. Comparison of the present results with other similar findings in the literature should be discussed in more detail. This is necessary in order to place this work together with other work in the field and to give more credibility to the present results.
10) Conclusion part is very long. Make it brief and improve by adding the results of your studies.
11) There are many grammatic mistakes. Improve the English grammar of the manuscript.
Author Response
Please see the attached response.

Reviewer 3 Report
The paper entitled " Kuhn verdazyls: new insights into the synthesis and mechanism of formation” describes the optimization of synthesis protocol of Kuhn verdazyls from arenediazonium salts, hydrazones and formalin.
In my opinion, this article should be published in the International Journal of Molecular Sciences after minor revision:
1. I strongly suggest to add copies of IR spectra to supporting information.
2. Add peak intensities to the IR spectra absorptions lists in the experimental part.
3. In the experimental part it is indicated that TLC was used for analysis. Therefore it would be useful to add for each substance Rf values and eluent
Author Response
Please see the attached response.

Round 2
Reviewer 1 Report
I have read the corrected version and I have no objection to the reaction mechanism proposed. However, lines 149-150 contain the shocking news that the free radical in an acidic environment can becomes a VDC cation. From the point of view of chemistry, this is not acceptable. The measurement of the ESR spectrum of an acidified 3a sample should not show the presence of a free radical, but this probably does not happen. In the described case, the nitrogen atom adjacent to the atom on which there is a free radical is protonated. And this causes the observed changes.
Author Response
The authors are grateful to the reviewer for their valuable and positive comments on our manuscript. We did all our best to consider all the points raised by the referee and the responses are provided below.
Indeed, this was a mistake, we corrected the text according to the reviewer's comments, please see the revised version.